# Dense2MoE: Unifying Pruning and Upcycling for Efficient Large Language Models

## Abstract

The Mixture of Experts (MoE) architecture has become a mainstream design in Large Language Models (LLMs) for its ability to flexibly scale parameters while maintaining inference efficiency. However, training MoE models from scratch remains prohibitively expensive due to their high computational demands. Existing upcycling methods reduce costs by converting dense LLMs into MoEs through layer duplication and fine-tuning, but introduce substantial redundancy. Layer-wise pruning is commonly used to alleviate redundancy among the decoder layers of dense models, but it inevitably incurs performance degradation. In this paper, we propose Dense2MoE, a novel approach that unifies layer pruning and upcycling through a technique we term Layer-Fusion UpCycling(LF-UC). Our method prunes highly redundant layers in an LLM while retaining their MLPs in the form of MoE. In this way, tokens are routed through a subset of redundant MLP layers rather than all of them. This design efficiently leverages open-source LLMs with low additional computational overhead, enhancing model performance while reducing active parameters. Extensive experiments show that Dense2MoE effectively pushes the Pareto frontier of efficiency versus performance toward a more optimal region compared with the original seed models, and achieves a superior trade-off relative to alternative approaches.

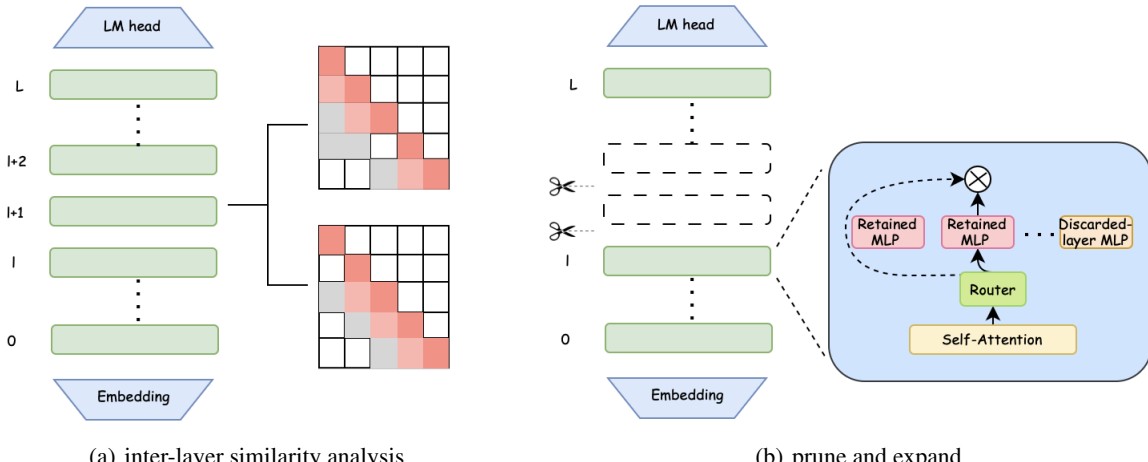

(a) inter-layer similarity analysis        (b) prune and expand

Figure 1: **Overview of Dense2MoE.** (a) inter-layer similarity analysis: we analyze the output similarity of each decoder layer in the LLM and the similarity of inputs to each Multi-Layer Perceptron (MLP). (b) Pruning and Layer-Fusion Upcycling (LF-UC): We prune the attention modules of layers identified as highly redundant based on the aforementioned similarity metrics. Instead of discarding the MLPs from these pruned layers, we fuse them into the MLP layers of the retained transformer blocks.

## 1 Introduction

The scaling law(Kaplan et al. (2020)) of large language models (LLMs) has established that as the model size and training dataset size continue to increase, the model's capabilities improve consistently. Consequently, a growing body of open-source work(Yang et al. (2025); Touvron et al. (2023); Jung et al. (2010); Tripti R et al. (2025)) has focused

on developing models with larger parameter sizes. However, the continuous expansion of model scale imposes higher requirements on hardware and poses challenges for deployment in resource-constrained hardware environments in real-world scenarios.

To more fully leverage the knowledge stored in open-source models and reduce resource waste caused by model retraining, many studies (Gromov et al. (2024); Yang et al. (2024); Ashkboos et al. (2024); Men et al. (2024); Song et al. (2024); Kim et al. (2024); Chen et al. (2024)) have opted to prune large-parameter models. These techniques remove redundant parameters or components in the model and can often ensure near-optimal performance in specific task domains. Another approach is parameter expansion, which increases the model's parameter count by extending the number of model layers or adopting the Mixture of Experts (MoE)(Shazeer et al. (2017)) framework—typically trading a small increase in active parameters for better model performance.

Nevertheless, existing methods have inherent limitations: Unstructured model pruning requires targeted operator optimization to accelerate sparse matrix computations, while structured pruning often leads to performance degradation due to the loss of critical model parameters. Meanwhile, parameter expansion methods (Komatsuzaki et al. (2022); Sukhbaatar et al. (2024); Wu et al. (2024)) increase the model's total parameter count, which in turn causes a multiplicative growth in redundant parameters.

To balance model performance and efficiency, we propose Dense2MoE—a novel method that jointly combines model pruning and parameter expansion. It enables open-source models to deliver better performance across diverse hardware platforms at extremely low training costs. An overview of Dense2MoE is illustrated in Fig.1: First, we analyze the output similarity between different decoder layers of the LLM and the similarity of outputs from self-attention modules. Based on this analysis, we prune highly redundant layers while retaining their MLPs to serve as experts within an MoE framework. Specifically, through our Layer-Fusion Upcycling (LF-UC) technique, we expand the MLPs of the retained layers into MoE structures, incorporating the upcycled MLPs as additional experts. Finally, we perform a small amount of finetuning to adapt the modified model structure and parameters.

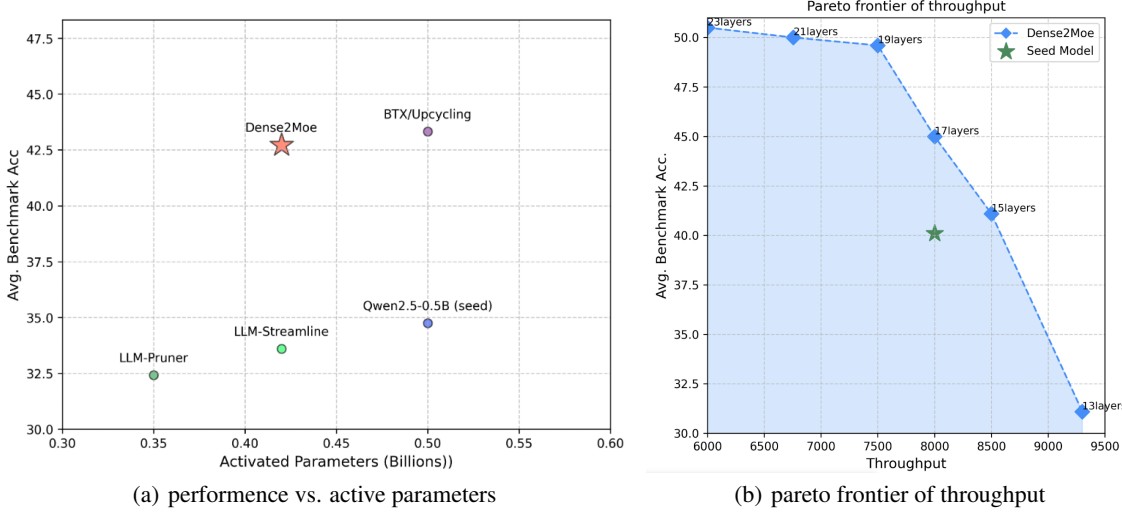

(a) performance vs. active parameters        (b) pareto frontier of throughput

Figure 2: **Performance Comparison with the Seed Model and Alternative Methods.** (a) performance vs. active parameters mathematics, code, reasoning and general knowledge benchmarks, among different pruning methods and parameter upcycling. (b) pareto frontier of throughput

Furthermore, most existing model pruning works evaluate accuracy based on metrics such as perplexity (PPL), validation loss, or benchmarks in general knowledge. In contrast, we comprehensively evaluate our Dense2MoE across benchmarks spanning mathematics, code, and general knowledge domains. This multi-domain assessment fully validates the practical applicability of our method in real-world scenarios.

In summary, our key contributions are as follows:

- We propose a simple yet novel framework for leveraging open-source models, improving the Pareto frontier of efficiency versus accuracy when compared with seed models. Dense2MoE achieved better performance with fewer activated parameters.

- We conduct comprehensive evaluations across benchmarks in general knowledge, code, and mathematics domains. The results demonstrate that our method achieves an optimal balance among approaches focused on model structured pruning and model expansion.

- We validate the scalability of our method across open-source models of different scales and model families.

## 2 METHOD

Dense2MoE adopt a prune-and-expand approach to reuse open-source models and transform them into MoE, aiming to reduce active parameters while improving performanc. As illustrated in Fig.1. Our method can be summarized into two key steps: (1) identify layers where both the decoder layer outputs and MLP inputs are similar; (2) retain the first layer that meets the condition in (1), drop all subsequent layers, and retain their MLPs in the first layer as experts. After these two steps, we perform fine-tuning on a small volume of data for the modified model.

### 2.1 PRELIMINARY

**Forward Propagation in LLMs.** Modern decoder-only large language models (LLMs), such as GPT, primarily stack transformer blocks—composed of attention mechanisms and multi-layer perceptron (MLP) layers—along the depth dimension. Specifically, the inter-layer information transmission in the transformer architecture is given by a residual iteration equation

$$x^{(l+1)} = x^{(l)} + f(x^{(l)}, \theta^{(l)}) \tag{1}$$

Where $x^{(l)}$ denotes the hidden state input to the $i$-$th$ transformer block, and $\theta^{(l)}$ represents the parameters of the $i$-$th$ layer. The $i$-$th$ layer function f contributes a transformation of $f(x, \theta)$ to $x^{(l)}$, which consists of a multi-head attention (MHA) layer and a MLP layer

$$f(x^{(l)}, \theta^{(l)}) = \text{MHA}^{(l)}(x^{(l)}, \theta_{mha}^{(l)}) + \text{MLP}^{(l)}(h^{(l)}, \theta_{mlp}^{(l)}) \tag{2}$$

Where, $h^{(l)}$ denotes the input to the MLP

$$h^{(l)} = x^{(l)} + \text{MHA}^{(l)}(x^{(l)}, \theta_{mha}^{(l)}) \tag{3}$$

**Mixture of Experts (MoE).** Mixture of Experts models expand the model parameter count by replacing the MLPs in traditional Transformers with MoE layers. Owing to their sparse activation property, MoE models can typically achieve better performance than dense models while activating fewer parameters.

Each MoE layer comprises a router which assigns the input token representation $h^{(l)}$ to N MLP experts via a router network, parameterized by $W_g \in \mathbb{R}^{d_{\text{model}} \times N}$. The output of the MoE layer is computed as the weighted sum of outputs from the top-k experts with the highest probabilities assigned by the router. Denoting the set of selected top-k indices as K

$$\text{MLP}_{\text{MoE}}^l = \sum_{i \in K} g_i(h^{(l)}) * \text{MLP}^{(i)}(h^{(l)}) g_i(h^{(l)}) = \text{softmax}^{(i)}(h^{(l)} W_g) \tag{4}$$

### 2.2 INTER-LAYER SIMILARITY ANALYSIS

Analyzing the similarity between transformer layers is crucial for our pruning-and-expansion method. We use cosine similarity to measure the similarity between hidden states. The similarity between hidden state X1 and hidden state X2 can be expressed as

$$s(\mathbf{X_1}, \mathbf{X_2}) = \frac{\mathbf{X_1} \cdot \mathbf{X_2}}{\|\mathbf{X_1}\| \cdot \|\mathbf{X_2}\|} \tag{4}$$

and the distance $d(\mathbf{h^{(1)}}, \mathbf{h^{(1+n)}})$ between the MLP input $h^{(l)}$ of the l-th layer and the MLP input $h^{(l+n)}$ of the l+n-th layer.

$$s(\mathbf{h^{(1)}}, \mathbf{h^{(1+n)}}) = \frac{\mathbf{h^{(1)}} \cdot \mathbf{h^{(1+n)}}}{\|\mathbf{h^{(1)}}\| \cdot \|\mathbf{h^{(1+n)}}\|} \tag{5}$$

In this way, we obtain two distance lists $S_L = \{s(\mathbf{x}^{(l)}, \mathbf{x}^{(l+n)}) \mid l \in \{0, 1, \ldots, N-2\}, n \in \{1, 2, \ldots, N-1\}\}$ and $S_M = \{s(\mathbf{h}^{(l)}, \mathbf{h}^{(l+n)}) \mid l \in \{0, 1, \ldots, N-2\}, n \in \{1, 2, \ldots, N-1\}\}$. Similar to many previous works, we hypothesize that layers with greater inter-layer output similarity $s(\mathbf{x}^{(l)}, \mathbf{x}^{(l+n)})$ exhibit higher redundancy. However, instead of pruning all components of these layers as prior methods did, we retain their MLPs—the components more

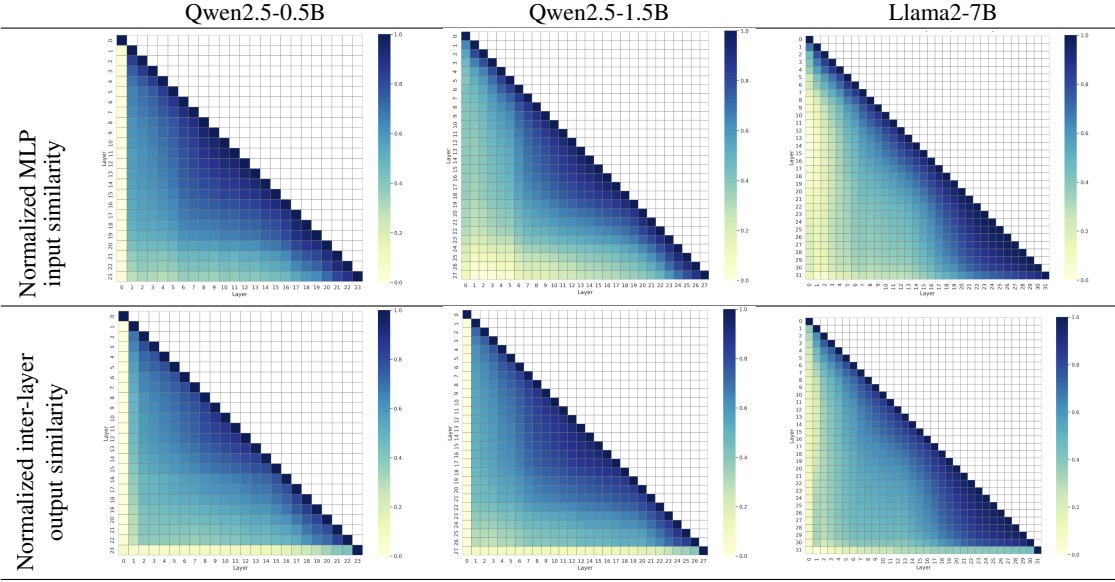

Figure 3: Normalized similarity comparison across different models.

critical to model performance. For layers with high MLP input similarity $s(\mathbf{h}^{(\mathbf{l})}, \mathbf{h}^{(\mathbf{l+n})})$, we adopt a strategy of sharing attention layers (Section 2.3).

We visualize $S_L$ and $S_M$ across all layers for Qwen2.5-0.5B, Qwen2.5-1.5B, and LLaMA2-7B. These similarity metrics serve as prior knowledge for our pruning-and-expansion approach. In Fig.3, the color of each grid cell represents the distance between the hidden states of different layers. Layers with greater similarity exhibit relatively higher redundancy. We perform pruning-and-expansion at appropriate positions with reference to these similarities.

## 2.3 PRUNING AND EXPANSION

After the detailed analysis in Section 2.2, we obtain a set of output similarities $S_L$ for each decoder layer and a set of similarities $S_M$ for each MLP input. We consider that decoder layers with higher output similarity $s(\mathbf{x}^{(\mathbf{l})}, \mathbf{x}^{(\mathbf{l+n})})$ have higher redundancy, an idea that has also been confirmed by Gromov et al. (2024). Inspired by previous studies(He et al. (2024)), where MLP layers serve as knowledge repositories and pruning MLP layers has a significant impact on performance, while attention layers exhibit higher redundancy. We prune layers in $S_L$ that have high values and also show approximate MLP inputs (i.e., $s(\mathbf{h}^{(\mathbf{l})}, \mathbf{h}^{(\mathbf{l+n})})$). By sharing the attention mechanism from previous layers, the MLPs of redundant layers are retained as optional paths (Fig. 4).

Specifically, we sort $S_L$ in descending order and search for $l^*$ and $n^*$ in $S_L$ that satisfy $s(\mathbf{h}^{(\mathbf{l})}, \mathbf{h}^{(\mathbf{l+n})}) < \delta$, where $\delta$ is a hyperparameter. We expand the parameters in the form of MoE: we replicate the MLP of layer $l^*$ N times to serve as N experts, and replicate the MLPs from layer $l^*$ to $l^* + n^*$ M times each, incorporating these $n^* * M$ experts into layer $l^*$. That is,

$$x^{(l^*+n^*)} = x^{l^*-1} + \text{MHA}^{(l^*)}(x^{(l^*-1)}) + \sum_{i=1}^{K} \alpha_i * \text{MLP}^{(l^*)}(h^{(l^*)}) + \sum_{k=1}^{n^*}\sum_{j=1}^{M} \beta_j * \text{MLP}^{(l^*+k)}(h^{(l^*)}) \quad (6)$$

Evidently, our method prunes the attention layers from layer $l^* + 1$ to $l + n^*$ and shares the hidden state $h^{(l^*)}$ of the MLP input at layer $l^*$. Due to the multi-head and redundant nature of the attention mechanism, pruning attention layers with high similarity does not result in significant performance degradationHe et al. (2024). Since we ensure the condition $s(\mathbf{h}^{(\mathbf{l})}, \mathbf{h}^{(\mathbf{l+n})}) < \delta$ during merging, the MLPs from layer $l^* + 1$ to $l + n^*$, when serving as experts, can restore the previous outputs as much as possible.

Our method decouples various structures within the main path of the LLM, enabling the model to learn and select network structures via residual paths. This approach mitigates redundancy arising from the serial arrangement of model layers and significantly enhances the scalability of the model's parameter count through replication.

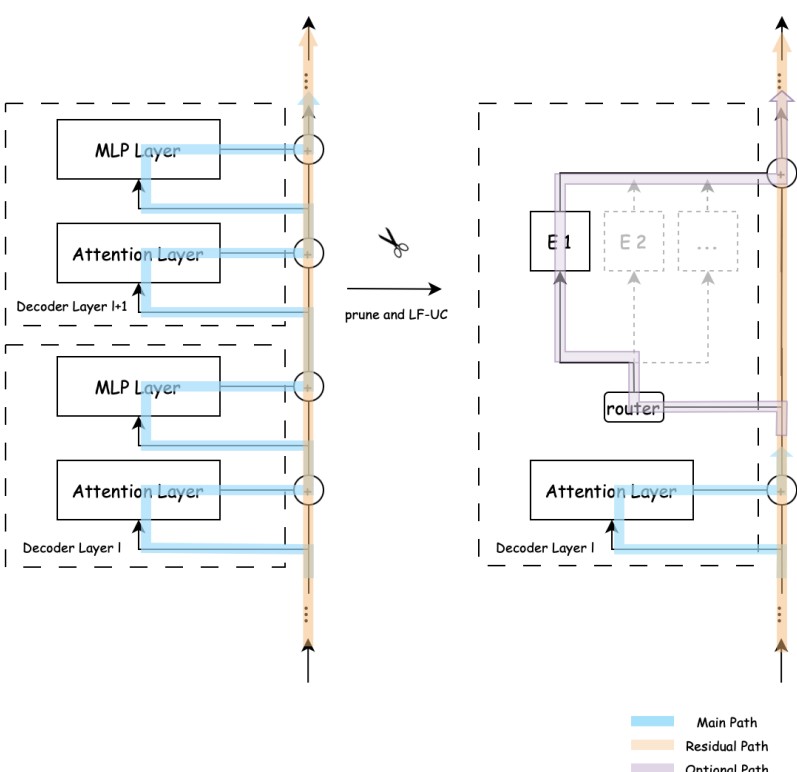

Figure 4: Tokens skip redundant layers through path selection.

## 3 EXPERIMENTS

We validated the Dense2MoE method on models of different scales across two open-source model families (Qwen2.5Team (2024) and Llama2Touvron et al. (2023)), and compared it with prevalent model layer pruning and upcycling approaches. For fairness, all methods were fine-tuned on a small-scale dataset of equal size, and their performance was validated on benchmarks spanning multiple domains (mathematics, code, reasoning and general knowledge).

### 3.1 SETUP

Considering the application scenarios of edge-side models, we conducted our experiments on Qwen2.5-0.5B(Team (2024)). To verify the generality of our method, we further extended it to models of larger scales and different families (Qwen2.5-1.5B(Team (2024)), Llama2-7B(Touvron et al. (2023))). Unless otherwise specified, our experimental settings are as follows: the modified model was trained with Continual Pre-Training on a dataset of 180B tokens, using a batch size corresponding to 40 million tokens. For the learning rate schedule, we implemented a warm-up phase from 0 to 1e-4 over 360 million tokens, followed by a cosine decay to 1e-5 after reaching the peak value.

In addition, we constructed a 225B-token dataset with the same composition to compare model performance under the same training resource consumption as the upcycling approach.

We compared our model with several mainstream works on model layer pruning and parameter expansion across multiple benchmarks. Specifically, for the general knowledge domain, we validated the superiority of our method on C-Eval(Huang et al. (2023)), CMMLU(Li et al. (2023)), and MMLU(Hendrycks et al. (2020)). To compare the mathematical and coding capabilities of different methods, we conducted evaluations on GSM8K(Liu et al. (2023)), CMath(Wei et al. (2023)) (mathematics), HumanEval(Chen et al. (2021)) and MBPP(Austin et al. (2021)) (coding). For assessing the model's reasoning ability, we adopted BBH(Suzgun et al. (2022)) and ARC-Challenge(Clark et al. (2018)).

Table 1: Domain data sources and composition.

| Domain | Dataset | Sampling ratio (%) |
|---|---|---|
| | OpenWebMath | 12.0 |
| | Arxiv | 14.0 |
| Math | Github | 2.3 |
| | Code | 16.2 |
| Code | Synthetic Data | 3.8 |
| | Wikipedia | 50.6 |
| General Knowledge | COIG | 6.8 |
| | C4 | 2.6 |

## 3.2 DATASET DETAILS

Our dataset is primarily assembled from publicly available, open-source sources, which we meticulously cleaned, normalized, and deduplicated to ensure high data quality. After processing, the dataset comprises approximately 180 billion tokens, which serve as the foundation for fine-tuning our models. To better align the Dense2MoE-modified architecture with diverse downstream tasks, we carefully adjusted the proportions of data from different domains, thereby enabling targeted re-adaptation of the model parameters. This approach ensures that each domain is adequately represented during fine-tuning, improving both generalization and task-specific performance. A detailed breakdown of the dataset composition, including the domain-specific proportions and token counts, is provided in Table1.

## 3.3 MAIN RESULTS

**Compared with the seed model, Dense2MoE advances the Pareto frontier of performance versus throughput.** We measured throughput on a single NVIDIA A800 GPU (80GB HBM2e VRAM) with a batch size of 32, a prompt length of 1023, and a maximum generation length of 256 tokens. As shown in Fig.2(b), under the same settings, Dense2MoE achieves a superior efficiency-performance trade-off after pruning different numbers of decoder layers. At comparable throughput, Dense2MoE delivers a 12.5% performance improvement (40.1 → 45).

Table 2: Performance comparison of different methods across various benchmarks

| Method | Activated Parameters | Benchmarks | | | | | | | | | Avg. |
|---|---|---|---|---|---|---|---|---|---|---|---|
| | | CEVAL | CMMLU | MMLU | GSM8K | CMath | HUMANEVAL | MBPP | BBH | ARC-C | |
| Qwen2.5-0.5B | 0.5B | 53.77 | 52.06 | 47.6 | 38.59 | 30.66 | 28.66 | 29.6 | 29.85 | 26.96 | 37.53 |
| UIDL | 0.39B | 43.63 | 45.28 | 36.1 | 26.54 | 30.5 | 27.44 | 21.8 | 27.5 | 23.29 | 32.35 |
| LLM-Pruner | 0.4B | 42.87 | 44.79 | 37.2 | 34.7 | 34.59 | 24.13 | 22.6 | 26.43 | 24.5 | 32.42 |
| LLM-Streamline | 0.4B | 45.75 | 45.94 | 47.62 | 34.17 | 28.51 | 25.63 | 20.76 | 29.2 | 26.34 | 33.67 |
| Dense2MoE | 0.4B | 59.68 | 57.8 | 44.7 | 43.37 | 43 | 36.59 | 41 | 33.79 | 24.57 | **42.72** |

**Compared with related pruning and upcycling methods, Dense2MoE achieves superior performance across multiple benchmarks.** We conducted a comprehensive evaluation using 9 benchmarks covering general knowledge, mathematics, code, and reasoning. As shown in Table 2, after training on 270B tokens, Dense2MoE activates only 80% of the seed model's parameters while improving average performance by more than 7 percentage points. Furthermore, under the same training FLOPs, Dense2MoE outperforms upcycling methods with larger parameter counts (Table3).

Table 3: Performance comparison between Dense2MoE and Upcycling under equal FLOPs

| Method | Activated Parameters | Trained Tokens | Avg. Acc |
|---|---|---|---|
| Upcycling | 0.5B | 180B | 46.24 |
| Dense2MoE | 0.4B | 225B | **48.17** |

**Generalization and scalability analysis.** To validate the generality and scalability of our method, we extend Dense2MoE to models of different families and sizes, including LLaMA2-7B and Qwen2.5-1.5B. The results are

presented in Table 4. After fine-tuning Dense2MoE using only 180B tokens, it outperforms Qwen2.5-0.5B by 3.04 percentage points and LLaMA2-7B by 11.16 percentage points, demonstrating that our model remains effective as model scale increases and can be extended to other open-source model families.

Table 4: Performance comparison of different methods across various benchmarks

| LLM | Model | Activated Parameters | Benchmarks | | | | | | | | | Avg. |
|---|---|---|---|---|---|---|---|---|---|---|---|---|
| | | | CEVAL | CMMLU | MMLU | GSM8K | CMath | HUMANEVAL | MBPP | BBH | ARC-C | |
| Qwen2.5 | Dense | 1.5B | 68.72 | 67.82 | 61.1 | 49 | 63.99 | 35.98 | 46 | 39.73 | 64.42 | 55.19 |
| | Ours | 1.2B | 72.51 | 70.5 | 55.5 | 51.67 | 63.31 | 52.44 | 53.2 | 42.73 | 62.2 | **58.23** |
| Llama2 | Dense | 7B | 30.99 | 32.75 | 45.8 | 22.83 | 16.6 | 12.8 | 21.6 | 39.36 | 27.8 | 27.84 |
| | Ours | 5.7B | 59.43 | 51.2 | 39.8 | 34.82 | 24.8 | 32.5 | 29.94 | 36.43 | 42.1 | **39** |

## 3.4 ABLATION STUDIES

**Layer-Fusion Upcycling (LF-UC) outperforms the naive combination of redundant layer pruning and upcycling.** In Table 5, we compare our LF-UC with a baseline approach that prunes layers based on inter-layer similarity as the redundancy criterion, then simply duplicates and expands the retained layers' MLPs into MoEs through upcycling. Experimental results demonstrate that our method achieves superior performance, confirming that Dense2MoE better preserves LLM's original capabilities while eliminating redundancy.

Table 5: Performance comparison of different methods across various benchmarks

| Method | Benchmarks | | | | | | | | | Avg. |
|---|---|---|---|---|---|---|---|---|---|---|
| | CEVAL | CMMLU | MMLU | GSM8K | CMath | HUMANEVAL | MBPP | BBH | ARC-C | |
| Prune+Upcycling | 53.72 | 54.92 | 41.8 | 39.67 | 38.06 | 36.58 | 37.8 | 26.73 | 27.05 | 39.59 |
| Prune+Layer-Fusion Upcycling | 59.68 | 57.8 | 44.7 | 43.37 | 43 | 36.59 | 41 | 33.79 | 24.57 | **42.72** |

## 4 ANALYSIS

### 4.1 SCALING ANALYSIS OF THE NUMBER OF PRUNED LAYERS

We empirically searched for the impact of the number of pruned layers on model performance, as shown in Figure 5. We pruned the top-k redundant layers($k = 1, 3, 5, 7, 9, ...$) and expanded the parameters via Layer-Fusion Upcycling (LF-UC). The results demonstrate that Dense2MoE effectively mitigates pruning-induced performance degradation when the number of pruned layers is within a certain range ($k \leq 8$). However, when the number of pruned layers exceeds a threshold, the performance of Dense2MoE degrades rapidly, as the negative impact of layer-wise pruning becomes the dominant factor.

### 4.2 SCALING ANALYSIS OF THE NUMBER OF EXPERTS

We scaled the total number of experts per layer. Under the top-1 gating setting, model performance improves sharply as the number of experts increases, but the improvement slows when the number of experts reaches a certain point, limited by the scale of training data and active parameters. Thus, for Dense2MoE at the Qwen2.5-0.5B scale, we chose a configuration with a total of N=6 experts and pruning the top-5 redundant layers.

## 5 RELATED WORK

The Mixture of Experts (MoE) framework has emerged as a dominant paradigm for scaling large language models (LLMs) while balancing computational efficiency, addressing the limitations of dense models where parameter growth linearly increases inference costs. By decomposing the model into specialized "expert" sub-modules and a routing mechanism that activates only a subset of experts per token, MoE achieves parameter sparsity without sacrificing capacity.

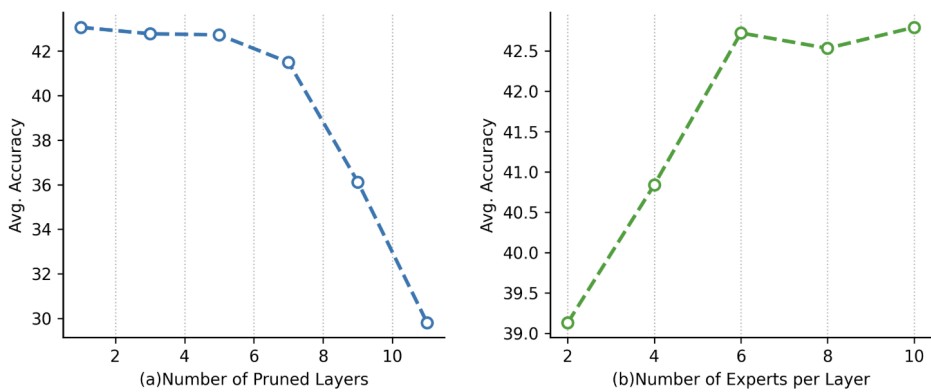

Figure 5: Scaling Analysis of Layers and Experts

However, the computational cost of training MoE models from scratch remains prohibitive, prompting research into efficiently leveraging open-source models for resource reuse. Model pruning is a straightforward approach that calculates similarity between weights, considers dimensionality reduction, and evaluates parameter activation frequency to eliminate redundancy. Unstructured pruning maintains performance while sparsifying the model but creates irregular parameter patterns that often require operator-level customization and optimization. In contrast, structured pruning ensures parameter regularity and better compatibility but risks significant performance degradation by removing entire neurons or layers, as it deletes larger, potentially more critical components.

Upcycling methods extend dense model parameters by reusing attention mechanisms and replicating MLPs, cost-effectively improving performance under equivalent activated parameter constraints. However, this approach increases the redundancy already present in LLMs.

Our Dense2MoE integrates layer-wise pruning with upcycling through our proposed Layer-Fusion Upcycling (LF-UC) technique, removing redundant LLM layers while preserving their more critical MLP components. This approach enables Dense2MoE to achieve a superior balance between performance and efficiency compared to standalone pruning or upcycling methods.

# 6   CONCLUSION

In this work, we propose Dense2MoE, a novel framework that jointly combines layer-wise pruning and parameter upcycling to enhance both efficiency and performance of large language models. By leveraging the Layer-Fusion Upcycling (LF-UC) technique, Dense2MoE identifies and removes redundant layers while repurposing valuable MLP parameters as additional experts in an MoE structure, achieving a balance between parameter efficiency and model capacity. Comprehensive evaluations across multiple domains, including mathematics, code, and general knowledge, demonstrate that Dense2MoE consistently outperforms approaches relying solely on pruning or parameter expansion. Moreover, the method generalizes effectively across different model families and scales, enabling practical deployment on diverse hardware with minimal training costs.

Dense2MoE highlights a synergistic strategy for optimizing open-source models, paving the way for further exploration of expert activation, attention upcycling, and integration with other compression techniques to maximize efficiency without compromising performance.

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
