# OpenReview forum: "Dense2MoE: Unifying Pruning and Upcycling for Efficient Large Language Models"
_ICLR.cc/2026/Conference — Submitted to ICLR 2026_

### Official Review · Reviewer_Y7cZ · 2025-10-29

**Soundness:** 2
**Presentation:** 1
**Contribution:** 2
**Rating:** 2
**Confidence:** 5

**Summary:**

The paper proposes Dense2MoE, a recipe that converts a dense LLM into a sparse MoE model while reducing active parameters. The key idea, Layer-Fusion UpCycling first measures inter-layer output similarity and MLP-input similarity to find redundant blocks. It then prunes the attention of these redundant layers but retains their MLPs as experts fused into earlier surviving layers, with lightweight fine-tuning afterward. Experiments report a better accuracy–throughput Pareto than the seed models and improvements over pruning and upcycling baselines across multiple benchmarks.

**Strengths:**

1- Simple recipe that can be replicated easily.

2- Comprehensive evaluation on multiple benchmarks.

3- Experiments on multiple model scales and model families.

**Weaknesses:**

1- The manuscript needs substantial editorial work. It contains frequent typos, introduces concepts without definition, and offers several unclear explanations. Overall it reads like an early draft or work-in-progress rather than a polished submission.

2- The claim of “low additional computational overhead” is incompatible the reported continual pre-training budget of 180B–225B tokens.

3- The related-work section lacks citations and context, and the experimental baselines are under-specified. Important comparators such as Llama-MoE [1] and ToMoE [2] are not cited or evaluated.

4- In Table 2, it's unclear what the baselines are. Are the baselines and the dense model also trained using the same amount of compute as the proposed method? If not the performance improvements over the dense model might be as a result of this training and overlap between the continual pre-training dataset and the benchmarks. I suggest the authors try CPT on Wikitext data and report the results.

5- It's unclear what the method introduced as upcycling in table 3 is. It is trained with fewer tokens.

6- How are hyper-parameters selected? what are the values of K and M? How is $\delta$ set?

[1] LLaMA-MoE: Building Mixture-of-Experts from LLaMA with Continual Pre-training, Zhu et al, 2024

[2] ToMoE: Converting Dense Large Language Models to Mixture-of-Experts through Dynamic Structural Pruning, Gao et al, 2025

**Questions:**

1- Figure 1 caption: The caption should clarify why two MLPs are retained. This rationale only becomes clear after reading the method section. It could be briefly summarized in the caption for clarity.

2- Lines 118–119: Revise to make it explicit that only the first layer in the identified set is retained, while the remaining layers are removed. The current phrasing implies that all subsequent layers are discarded.

3- Line 154: The text introduces the distance $d$, but the subsequent equation again defines a similarity measure. Please reconcile or clarify the intended relationship between the two.

4- Line 160: The phrase “similar to prior works” should be supported by specific citations. Please add the relevant references.

5- Line 200: The description mentions searching for $l^*$ and $h^*$, yet the following equation uses $h$ and $l$. Clarify how $l^*$ and $h^*$ are calculated.

6- Line 211: Typo error in the citation command.

7- Equation 6: Define all symbols clearly. What does $K$ represent? is it equivalent to $N$? What are $\alpha$ and $\beta$ Why do all replicated experts share the same coefficients?

8- Router definition: The router mechanism is not introduced or described. Please include its formulation and role in routing tokens to experts.

9- Unnumbered equation between Eq. 3 and Eq. 4:  This equation should either be numbered. It should be revised for clarity.

10- Figure 4: The figure occupies a large portion of the page but provides little additional information beyond Figure 1(b). Consider removing it.

11- Line 248: Correct citation formatting errors.

12- Line 194 :The set $S_M$ is never used later in the text. Please clarify its role or remove it for consistency.

Overall the paper needs a major revision.

---

### Official Review · Reviewer_2ery · 2025-11-01

**Soundness:** 2
**Presentation:** 3
**Contribution:** 2
**Rating:** 2
**Confidence:** 4

**Summary:**

The paper proposes Dense2MoE, a method that unifies layer pruning and parameter upcycling to efficiently transform dense LLMs into Mixture-of-Experts (MoE) models. Instead of discarding redundant layers entirely, the approach prunes their attention modules but reuses the MLP submodules as experts within retained layers through a mechanism called Layer-Fusion UpCycling (LF-UC). This reduces active parameters while maintaining performance. The method is validated on models such as Qwen2.5 and LLaMA-2, showing better accuracy-efficiency trade-offs than existing pruning or upcycling baselines.

**Strengths:**

* The paper introduces a clear, conceptually simple mechanism that leverages layer redundancy for expert construction without extra pretraining costs

* Includes analysis on the number of pruned layers, experts, and scaling behavior, providing empirical justification for design choices

**Weaknesses:**

* The paper does not mention a very important thing: the total parameter counts. Results are framed in terms of activated parameters (e.g., Table 2), but MoE expansion increases total parameters; Comparing MoE and dense model with the same activated parameters count is not completely fair as MoE have much more param to store knowledge. This should be a thing that is made clear and well discussed.

* Throughput set-up may not reflect memory-bound regimes. Throughput is measured on a single A800 with fixed batch size; it does not explore max batch under memory constraints where larger total params could hurt throughput.

* The configuration of the experiments is not clearly specified. It would be beneficial to describe the detailed configuration of the model at the start of the experiment section or point to an appendix. I find the N=6 experts in the end of the experiments (section 4.2), which can be hard for reader to notice.

* It would be helpful to include more ablation studies. Section 3.4 is interesting, which compares LF-UC vs a prune+upcycle. While comparing performance of dense and moe model is not trivial, it would be more valuable to provide more comparison under fair setup to justify the design choices.

* There is no reference in the related work section. Also the baselines (e.g. LLM-streamline) is not properly introduced and cited.

**Questions:**

* Table 3 equalizes FLOPs by giving Dense2MoE more tokens (225B vs 180B); what will the results look like if they use the same number of tokens?
* How is the similarity threshold determined? Is it a sensitive hyperparameter?
* As the proposed method add additional training, is the seed model also further fine-tuned? If not, will the seed model benefit from that training?

---

### Official Review · Reviewer_kteM · 2025-11-03

**Soundness:** 3
**Presentation:** 3
**Contribution:** 3
**Rating:** 4
**Confidence:** 2

**Summary:**

This paper proposes Dense2MoE, a method to efficiently convert pretrained dense LLMs into mixture-of-experts (MoE) architectures without full retraining. The key idea is Layer-Fusion Upcycling (LF-UC), which identifies and prunes redundant transformer layers based on inter-layer similarity, and retains their MLP components as MoE experts. This allows the model to reuse valuable parameters while reducing redundancy and maintaining throughput efficiency. Empirically, Dense2MoE outperforms both pruning-only and upcycling-only baselines across multiple domains (math, code, reasoning, and general knowledge).

**Strengths:**

1. The motivation is solid - existing pruning hurts performance while upcycling increases redundancy, so combining them makes intuitive sense. The inter-layer similarity analysis is straightforward and the visualizations in Figure 3 effectively show where redundancy exists across different models.

2. I appreciate the comprehensive evaluation across multiple domains. The scalability experiments on different model families (Qwen2.5 and Llama2) and scales (0.5B to 7B) suggest the approach generalizes well, which is important for practical adoption.

**Weaknesses:**

My main concern is the lack of comparison with MoE models trained from scratch. This paper claims that Dense2MoE achieves efficiency “comparable to MoE” but does not compare to a real MoE baseline. Without this, it’s unclear whether Dense2MoE provides a meaningful alternative or merely a lightweight compression trick. The whole premise of “dense → MoE” upcycling is only interesting if the resulting model can approach the capability or efficiency of a true MoE system.

**Questions:**

1. How does Dense2MoE compare to a small MoE model trained from scratch with a similar number of active parameters?

2. The router initialization isn't mentioned. Are routers trained from scratch during fine-tuning? How is load balancing handled?

3. What happens if the inter-layer similarity threshold $\delta$ is mis-tuned? How sensitive are results to this choice?

---

### Official Review · Reviewer_BrSW · 2025-11-08

**Soundness:** 3
**Presentation:** 3
**Contribution:** 3
**Rating:** 6
**Confidence:** 3

**Summary:**

The paper proposes Dense2MoE, a unified layer pruning and upcycling framework that turns a dense LLM into a MoE by pruning redundant attention layers while fusing their MLPs as experts via Layer-Fusion UpCycling. WIth light fine-tuning, it claims better Pareto frontier of accuracy vs efficiency than original seed models and alternatives across math, code, reasoning and general knowledge questions.

**Strengths:**

- The paper proposes a simple but pratical method by pruning the attention layers and upcycling MLPs as experts.
- The paper conducts extensive experiments with a broad selection of benchmarks, and shows better average accuracy with 80% active parameters comparing to the seed model. It wins under equal FLOPs over upcycling methods with larger parameter counts.
- The paper is well presented. The pipeline and motivation are explained with similarity heatmaps and diagrams.

**Weaknesses:**

- The paper comes with limited analysis of routing stability and expert utilization. Consider to give training losses and gate temperature settings.
- $\delta$ threshold and search strategy for $l^*$ and $n^*$ are only described briefly without sensitivity analysis.
- Though LF-UC is compared with naive pruning + upcylcing, there are more that can be presented, e.g. prune-only, upcycle-only with the same tokens. Also, more analysis such as the effect of sharing attention vs not, and study how the optimal expert count scales across different base models and sizes since the best number of experts may change with model size and family.
- Only the single-GPU A800 numbers are presented. Need the latency and throughput under different batch number, which would be important for deployment.

**Questions:**

- How sensitive is performance to $\delta$ and to the number of experts per fused layer across different base models?

---

### Comment · Area_Chair_6Mu5 · 2025-11-25

Dear Reviewers,
Thank you to those who have already begun interacting with the authors — your timely follow-ups are greatly appreciated and reflect the professionalism and care that uphold our community’s standards.
For reviewers who have not yet responded, I would like to offer a gentle reminder. Authors have invested substantial time and effort into preparing their rebuttals, carefully addressing each concern raised in the reviews. As fellow researchers, we all understand the importance of being heard and having our clarifications considered. Even a brief acknowledgment or follow-up question helps ensure that the evaluation remains fair, thorough, and respectful of everyone’s work.
Your engagement during this phase is essential for maintaining a constructive and high-quality review process. Thank you again for your service and for treating both your fellow reviewers and the authors with the same consideration you would hope to receive.
Best regards,
AC

---

### Meta-Review · Area_Chair_QW6T · 2026-01-05

**Summary:**

The submission proposes Dense2MoE, a framework that unifies layer pruning and model upcycling. The method identifies redundant layers via similarity analysis, prunes the attention modules, and repurposes the MLP components as MoE experts in surviving layers.

The reviewers' primary concerns center on the lack of fairness in experimental comparisons, specifically the failure to evaluate against further-tuned dense baselines or MoEs trained from scratch despite a significant 180B-225B token training budget. Additionally, the paper lacks technical depth regarding total parameter counts, hardware throughput under memory constraints, and critical implementation details such as router initialization and hyperparameter sensitivity.

The authors' lack of engagement during the rebuttal phase, leaving these fundamental technical concerns unaddressed.

**Reviewer Concerns:**

The authors did not provide any responses or rebuttals to the reviewers’ comments. Consequently, no concerns were mitigated or clarified during the discussion period.

**Reviewer Scores:**

The authors' lack of engagement during the rebuttal phase, leaving these fundamental technical concerns unaddressed.

---

### Decision · Program_Chairs · 2026-01-26

Reject